# Effect of Midwife-Provided Orientation of Birth Companions on Maternal Anxiety and Coping during Labor: A Stepped Wedge Cluster Randomized Control Trial in Eastern Uganda

**DOI:** 10.3390/ijerph20021549

**Published:** 2023-01-14

**Authors:** Eva Wodeya Wanyenze, Gorrette K. Nalwadda, Josaphat K. Byamugisha, Patience A. Muwanguzi, Nazarius Mbona Tumwesigye

**Affiliations:** 1Department of Nursing, Mbarara University of Science and Technology, Mbarara 403, Uganda; 2Department of Nursing, College of Health Sciences, Makerere University, Kampala 101, Uganda; 3Department of Obstetrics and Gynecology, College of Health Sciences, Makerere University, Kampala 101, Uganda; 4Department of Epidemiology and Biostatistics, School of Public Health, Makerere University, Kampala 101, Uganda

**Keywords:** continuous support, outcomes, birth companion, low-resource setting, Uganda

## Abstract

The study aimed to assess the effect of midwife-provided orientation of birth companions on maternal anxiety and coping during labor. A stepped wedge cluster randomized trial design was conducted among 475 participants (control *n* = 240), intervention *n* = 235) from four clusters. Midwives in the intervention period provided an orientation session for the birth companions on supportive labor techniques. Coping was assessed throughout labor and anxiety scores were measured after birth. Independent t-test and Chi-Square tests were used to assess the differences by study period. Anxiety scores were reduced among women in the intervention period (*p* = 0.001). The proportion of women able to cope during early active labor was higher during the intervention period (*p* = 0.031). Women in the intervention period had 80% higher odds of coping (*p* = 0.032) compared to those in the control period. Notable differences in anxiety and coping with labor were observed among first-time mothers, younger women, and when siblings provided support. Midwife-provided orientation of birth companions on labor support lowers maternal anxiety and improves coping during labor. Findings could inform the planning and development of policies for the implementation of the presence of birth companions in similar low-resource settings.

## 1. Introduction

Improving the quality of care around the time of birth has been acknowledged as the most impactful strategy for reducing stillbirths and maternal and newborn deaths [1]. A woman’s emotional and cognitive experience of birth has a significant impact on her physical and psychological state. The process of labor comprises a great amount of physiological and psychological stress [2,3]. Women with moderate-to-high anxiety scores are more likely to experience poor progress in labor. Anxiety during childbirth contributes to dysfunctional uterine contractility by activating the release of stress hormones through the sympathetic nervous system [3,4,5]. Personal perceptions of labor and the degree of pain relief achieved can alter a woman’s sense of effective coping. The difficulties perceived by women regarding childbirth are closely related to their anxiety and the choice of coping methods. Helping a woman cope with the sensations she is experiencing may also help alleviate feelings of helplessness and suffering [6,7,8]. The World Health Organization (WHO) recommends that every woman be supported continuously throughout labor by a companion of choice [9,10]. Continuous labor support is defined as the presence of a trained professional or layperson or family members at the bedside of a parturient woman, to coach, empathize with, give practical aid to, and inform the expectant mother about birthing [11]. The companion in this context can be any person chosen by the woman to provide her with continuous support [9,10].

Labor companionship, like other non-clinical interventions, has not been regarded as a priority in many settings, yet it is an essential component of the experience of care [10]. Several barriers have been identified in the implementation of the presence of a companion of choice at birth in resource-strained hospital settings. Amongst these is the absence of clear communication with the companion about their role [12]. Companions’ lack of confidence and clarity about their role may lead to a sense of powerlessness when a woman is in pain [13,14]. For implementation purposes, WHO recommends that labor companions have an orientation session on supportive labor companionship techniques. This is to ensure that their presence is beneficial to both the woman and her health care providers [10,15].

Presently, women in Uganda are allowed to have a companion of choice during labor. These companions, however, neither receive an orientation nor have defined roles and responsibilities. It is acknowledged that actively involving family members in the process of labor ensures ownership and engagement [15,16].

There is low-quality evidence on the effect of continuous labor support in low-income settings. Additionally, it is still unknown whether training impacts the effectiveness of continuous labor support [17]. Training in this context is described as having an orientation session on supportive labor companionship techniques [10]. Continuous labor support is an important aspect of respectful maternity care. Findings from this study will add to the insufficient empirical evidence in low-income settings to inform policymakers on the implementation of the presence of birth companions. This study aimed to assess the effect of midwife-provided orientation on maternal anxiety and coping with labor. We hypothesized that midwife-provided orientation of birth companions on labor support techniques reduces maternal anxiety and aids coping during labor.

## 2. Materials and Methods

A cross-sectional stepped wedge cluster randomized trial was used. In this design, different individuals in the control and intervention are used with a single observation of outcomes [18]. This approach was selected because of the anticipated difficulty in simultaneously introducing the intervention to the different clusters. Additionally, it was preferred for ethical purposes; that is, not to withhold a beneficial intervention from some clusters. For purposes of this study, each selected facility was labeled as a cluster. The intervention was rolled out sequentially to the facilities over 12 months. The facilities were their controls, hence buffering the effects of heterogeneity between health facilities. In the first time block, all clusters were in the control phase and by the last time block, all clusters were in the intervention phase (see Figure 1) [18]. The trial was registered at the U.S. National Library of Medicine ClinicalTrials.gov trials on February 25, 2021 (NCT04771325). The reporting of this clinical trial is based on the CONSORT 2010 checklist [19]. 

The study was carried out in the Bugisu sub-region located in the eastern part of Uganda. The Bugisu sub-region consists of six districts, including Manafwa, Mbale, Bududa, Sironko, Namisindwa, and Bulambuli. According to the Uganda National Bureau of Statistics, the sub-region is home mainly to the Gisu people with an average household size of 4.8 and a literacy rate of 51.5% [20]. The sub-region has several health facilities, including one district hospital (Bududa) and one regional referral hospital in Mbale district. The Health Centre IVs (HCIV), district hospitals, and referral hospitals currently have a monthly average of 100, 200, and 600 deliveries, respectively. The HC IVs have an average of 12 midwives with one to two midwives per 8-h shift while the district and referral hospitals have about 18 midwives with two to three midwives per shift.

Cluster sampling: Each health facility in this context was considered a cluster. The inclusion criteria were the functionality of an operating theatre. Hospitals and HCIVs with a functional operating theatre were included. It was assumed that the presence of a functional theatre meant that the chances of referring women for cesarean sections to other facilities were small, enabling us to monitor more deliveries. Four clusters were selected for this trial. These were Mbale Regional Referral Hospital, Bududa, district hospital, Muyembe HCIV, and Busiu (Manafwa) HCIV.

Individual specific criteria. Women who had a birth companion, in spontaneously established labor and, expecting a vaginal delivery. Exclusion criteria were women with multiple pregnancies, previous cesarean section, and mental illness, or deaf or mute women. We excluded women who had previous cesarean sections because they had a higher chance of having another cesarean section.

Intervention: The intervention was “midwife-provided orientation of birth companions”. The admitting midwife provided an orientation session for the birth companion on supportive labor techniques. We assumed that providing an orientation session was likely to boost birth companion confidence, hence increasing the effectiveness of continuous support. The content for the orientation session consisted of providing emotional and physical support. Emotional support included being present, demonstrating a caring and positive attitude, saying calming verbal expressions, using humor, and praise, and encouraging and acknowledging efforts. Physical support included supporting her to change positions favoring upright positions, walking with her, giving her drinks and food, massaging, reminding her to go and pass urine, helping her find a comfortable position for pushing, and wiping her face with a cool cloth. The content was developed based on the literature on labor companionship techniques [10,11]. The orientation sessions were headed by four registered and licensed midwives. These directed and supervised the admitting midwives in the four respective clusters on the orientation of birth companions on labor supportive techniques. The first author EWW trained these midwives on how to conduct these sessions. Orientation was completed face-to-face, individually for the birth companion. This orientation was integrated into the admission procedure for the woman and lasted about 20 min. After labor was confirmed, in addition to the routine admission procedure, the midwife explained to the birth companion the different support techniques and clarified what was expected of them. Each task was explained in simple terms, including why the task was important and how it was performed, then companions were shown how it was performed, with return demonstrations from the companion. This was repeated for the birth companion to grasp and retain the task.

Control (usual care): Women are escorted to the health facilities by one or more family members or friends. One person is allowed, besides her, to provide support. The support persons do not receive any orientation sessions and have no designated roles. Routine analgesia is not given. Midwives, medical officers, and obstetricians provide skilled care. Typically, two to three midwives are allocated per 8-h shift managing about six laboring women at a given time.

Outcomes: This article is a part of a larger study assessing the effectiveness of midwife-provided orientation of birth companions on several outcomes. These were the incidence of having a spontaneous vaginal delivery, length of labor, Apgar score, coping, anxiety, and maternal satisfaction. The primary outcome was the chance of having a spontaneous vaginal delivery. This article is a reporting on two of the secondary outcomes of the actual trial. The rationale for reporting maternal anxiety and coping separately was to give more attention to the psychological aspects of childbirth. Psychological research on childbirth is scarce [21]. The authors found it necessary to deliberately disseminate these findings. This was to provide more of an account of the sociodemographic differences of women. Additionally, the trial was registered prior to completion of data collection to deter any bias in reporting of outcomes. Results presented in this manuscript are not selective nor post hoc positive findings, but are reported according to a specific study objective and the analysis plan was predetermined prior to data collection.

Sample size and randomization: Stepped wedge trials are designed to study the effect of an intervention [22]. The sample size for this trial was calculated based on the primary outcome (incidence of having a spontaneous vaginal delivery). The baseline rate for having a spontaneous delivery was 87%. We assumed that guiding birth companions on continuous labor support contributed to a 10% difference (minimally relevant difference between the two groups as 0.1). We set Alpha at 95% CI (0.05 = 1.96), β multiplier for 80% at 0.842. A sample size of 290 participants per period was calculated. The number of study participants was selected proportionately; that is, according to the patient volumes of the particular facilities. Approximately 12,500 women delivered during the study period including the 6 months break of the COVID-19 lockdown (see Figure 2). These also included women admitted in second stage, women who gave birth on their way to the hospital, elective cesarean sections, and complicated cases from lower health centers. In total, 580 eligible participants were recruited for the study. See details in Figure 3.

Randomization for stepped wedge trials is not performed individually but rather involves the crossover of clusters from control to intervention until all clusters are exposed [23]. Using a simple random technique, the principal investigator EWW generated a random sequence of the four hospitals. Numbers 1, 2, 3, and 4 were assigned to the different facilities (Mbale 1, Bududa 2, Muyembe 3, Manafwa 4). Using a random sequence generator, a sequence of “2, 4, 1, and 3” was generated. This sequence is what guided which facility crossed over first to the intervention period. Individual women who met the inclusion criteria were recruited from the clusters by study period. In cluster randomization trials, the intervention targets the cluster to prevent potential contamination of the control arm of study. There was a high chance of birth companions sharing what the midwives had shared with them with the other companions, leading to the need to randomize by cluster. Individual women were recruited because individual level outcomes were assessed [24,25].

Women’s self-reported anxiety levels were measured within 6 h after birth using the 10 cm Visual Analogue Scale for Anxiety (VAS-A). It ranges from 0 to 10, with a higher score representing high levels of anxiety. The VAS-A has demonstrated validity and reliability for measuring anxiety and has been used in several studies to assess anxiety in similar low-resource settings [26,27,28,29]. Coping with labor was assessed using the Roberts coping with labor algorithm [7]. The coping with labor algorithm was chosen because it emphasizes how well the laboring woman is coping with the physiologic changes and sensations. The main purpose of the algorithm is to not use the 0–10 numeric rating scale for pain in labor, because labor pain varies and is unique for individuals. According to the algorithm, the midwife looks out for cues for coping. These include being able to relax between contractions, rhythmic breathing, focus, moaning, counting, and rhythmic activity during contraction. Cues for not coping include crying, sweating, clawing, biting, and panicked activity during contraction [30]. This tool has been tested in large tertiary institutions with a variety of women from various ethnic backgrounds [31]. Coping was quantified as whether the woman is either coping well (CW) or not coping (NC). Coping was assessed at three periods, 4–7 cm; 8–10 cm, and second stage.

Analysis: In this study design, the distribution of results across control periods is compared with that across the intervention periods [32]. Anxiety and coping for women in the intervention period were compared to those in the control. Data were entered using excel and imported to STATA 14 for analysis. Participant baseline characteristics were summarized using frequencies and percentages. Anxiety was measured on a continuous scale with mean and corresponding standard deviation. Independent t-test was used to assess the difference in anxiety scores by study period. A *p*-value of <0.05 was taken to be statistically significant. Subgroup analysis was also undertaken to evaluate treatment effects for specific endpoint groups defined by specific baseline characteristics [33]. Multivariable analysis was also performed to assess the relative contributions of different factors that could affect maternal anxiety. Coping was assessed as a binary outcome; the proportion of those able to cope during the control and intervention period was determined. A Chi-Square test was used to test the difference in coping by study period. Prtest was also used to test the equality of proportions. Multivariable logistic regression was performed to assess potential confounders for the difference in coping. Both unadjusted and adjusted coefficients were presented together with the 95% confidence intervals and *p*-values.

## 3. Results

### 3.1. Participant Sociodemographic and Obstetric Characteristics

The majority of the respondents were in the age group of 15–24 years. Most of the respondents were first-time mothers 44.7%. The overall mean of gestation was 38.3 weeks (SD 1.0); Table 1 shows that the two groups were similar in many of the characteristics with no statistically significant difference except for marital status, support person, and cervical dilatation on admission. Statistical significance of these characteristics was coincidental since individual randomization was not performed; rather, it related to the order of introduction of the intervention to specific clusters.

Figure 3 shows selected clusters and the order of introduction of the intervention.

### 3.2. Maternal Anxiety

Maternal anxiety for women who received continuous labor support from birth companions who had a midwife-provided orientation (intervention) was compared to those who received usual care (control).

The overall mean anxiety score was 5.4 (SD = 2.0). Anxiety was higher in the pre-intervention period (5.7, SD = 1.9) compared to the post-intervention period (5.1, SD = 2.0). There was a statistically significant difference in the mean anxiety score between the pre-and post-intervention period (*p*-value = 0.001). See Figure 4.

Subgroup analysis to evaluate treatment effects for specific endpoint groups showed that although not statistically significant, there was a notable reduction in the mean anxiety score at the regional referral hospital (from (6.3-SD 2.3) to (3.4-SD2.2)). Additionally, high differences were noted among women who were younger, first-time mothers, with a low education status, and those who were married. Additionally, anxiety levels were much lower when sisters (siblings) (6.2(2.0) vs. 4.8(2.3)) offered support compared to the parents, spouses, friends, or other relatives. See Table 2.

A multivariable analysis was performed to assess the relative contributions of different factors that could affect maternal anxiety. We found a statistically significant difference in the anxiety scores by study period (*p* = 0.001). See Table 3.

### 3.3. Coping with Labor

The proportion of those able to cope was highest during the early active labor (89.7%) and lowest during the second stage (53.4%). A significant difference (*p* = 0.031) was found during early active labor (4–7 cm). These findings show that the percentage of women coping with labor reduced as the labor progressed. No significant findings were found for the later phases of labor. See Table 4.

Further assessment of coping at 4–7 cm within the clusters and other variables was performed to evaluate treatment effects for specific baseline characteristics. We found a statistically significant difference in the proportion of women coping by study period (*p* = <0.001) at the regional referral hospital. Furthermore, the intervention was more effective among those who were having their first child and the difference was statistically significant (*p* = 0.049). Though not statistically significant, the proportion of those able to cope was higher among the younger women (*p* = 0.07) and those with a lower level of education (*p* = 0.06) (see Table 5).

A multivariable analysis was performed to assess the relative contributions of different factors that could affect coping. Confounding baseline characteristics adjusted for were age, parity, cervical dilatation on admission, and relation of support person. Women in the intervention period had 80% higher odds of coping at 4–7 cm (Unadj.1.8; 95%CI 1.1–3.1) than those in the control period. The same results (Adj.1.8; CI 1.0–3.2) were found after adjusting. Furthermore, women having their second or more children and those who were supported by siblings had higher odds of coping. See Table 6.

## 4. Discussion

In this study, we assessed the effect of midwife-provided orientation for birth companions on maternal anxiety and coping with labor. Results from our study showed that maternal anxiety score was reduced and the proportion of women coping was higher in the intervention period. Similar findings are also reported in other related studies where the presence of trained husbands during delivery decreased maternal anxiety [34,35]. Findings from this study suggest that women may benefit more from continuous labor support when their birth companions are oriented by midwives on supportive techniques.

The current study further found that the intervention was more effective among first-time mothers. A study conducted in Malawi to determine the efficacy of a companion-integrated package for primigravid women showed that birth companions enhanced childbirth self-efficacy [36]. Studies have shown that primigravid women are more likely to experience anxiety compared to women who have given birth before [37,38,39]. Moreover, first-time mothers in a qualitative study acknowledged labor and childbirth as an unknown territory and were flexible with the process [40]. It is conceivable that women’s first experience of birth affects future reproductive health decisions, including the decision to have an institutional birth. The number of women having institutional births is still very low in the current study area. This finding may perhaps be used to advocate for emphasis on guided companionship as a supplementary intervention to address this issue. Furthermore, though not statistically significant, it is important to note the differences in the reduction in anxiety scores between the intervention and control periods across the different providers of supportive care. There was a greater difference in anxiety scores among those who received care from siblings compared to their spouses or mothers. An explanation for this could be that the mother or spouse was anxious and unable to help the women relax. An Iranian study also reported that some women did not want their mothers and husbands in the delivery room because they thought that they would be stressed by the labor pain [41]. This finding could be useful in tailoring orientations for the individual needs of companions for this recommendation to be more beneficial for women. Furthermore, the mean anxiety score was reduced at the regional referral hospital cluster while an increase was noted in the other three units. This could be attributed to the seniority of the midwives at the referral hospital. Moreover, this was in an urban setting and the effectiveness of the intervention was probably linked with income and educational level of birth companions [42]. It was probably easier for them to comprehend and execute given instructions from the midwives.

Coping during childbirth is a significant predictor of the development of post-traumatic stress disorder symptoms after birth [40,43]. Poor coping strategies are associated with anxiety and, in some cases, with negative health behaviours [8]. Non-pharmacological coping strategies can reduce labor pain and promote a satisfactory delivery experience [44,45]. Presently, there is little information on the effect of continuous labor support on coping during labor across all settings. In our study, there was a statistically significant difference in coping at 4–7 cm cervical dilation by study period. Our findings are similar to those from a study conducted to test the effectiveness of an educational intervention on coping. It was reported that the experimental group reported a significantly higher level of coping behavior than the control group [35]. Relatedly, a qualitative evidence synthesis on perceptions and experiences of labor companionship reported that a companion’s provision of emotional support through praise and reassurance increased women’s ability to cope. It was suggested that strong rapport and trust between the woman and companion led to women feeling more in control, enabling them to cope [46]. Components of labor support such as human reassurance and touch are known to facilitate the release of oxytocin which stimulates uterine myometrial contractions and other effects that counteract stress [47]. Nevertheless, positive findings of coping at 4–7 cm dilation in our study could partly be attributed to the fact that the majority of women came in early for admission during the intervention period. Having good supportive care may influence a woman’s perception of labor pain positively and her willingness to cooperate with given instructions. Emotional support from birth companions is vital for respectful maternity care and satisfaction following labor. Hence, we recommend that birth companions are directed on supportive techniques to mitigate the negative psychological outcomes of labor.

It is also imperative to note that the current study found non-significant findings on coping during the later stages of labor. A related study found that the anxiety levels of women were high during the last stage of labor, irrespective of the intervention [35]. A similar study also found that the late active phase pain scores were not significantly different [48]. Qualitative findings revealed that women tend to focus on themselves during the later stages and may not respond to instructions appropriately. Women described this phase as being in a world of their own and needing to be ‘alone’ [14,49]. The pain that precedes the actual birth of the baby has been described by the majority of women as unbearable. Thus, we advocate that birth companions are enlightened on the fragileness of these moments, and the need to stay calm to allow women to “be”, to reduce further distress. The current study additionally found that coping was also higher among primigravid mothers. An explanation for this finding could be that first-time mothers easily trusted the instructions of the birth companion as they had no previous birthing experiences to affect their response to pain. We, therefore, suggest that the presence of a birth companion be maximized to mitigate the potential intrapartum and postpartum risks for primigravid women.

Following WHO implementation, labor companionship has been implemented in Uganda to a certain extent. Birth companions, however, are not oriented nor the midwives trained on how to integrate the birth companion in the woman’s care. This study highlights the effectiveness of midwife-provided orientation on maternal anxiety and coping during labor. Findings could inform the feasibility of implementing the presence of birth companions. We would like to acknowledge that having 20 min for orientation of each birth companion in low-resource settings may perhaps be a challenge. This intervention can be modified by considering these options: group orientation sessions, conducting sessions during the antenatal period, and video recordings of supportive techniques playing on TV screens in the admission and waiting areas in busy facilities.

To our knowledge, this is the first study to assess the effect of continuous labor support on the events of labor and outcomes in Uganda. The selected study design enabled all the facilities to receive the intervention. The facilities were their controls, thereby buffering the effects of heterogeneity. Nonetheless, caution should be taken in generalizing the findings, given the following limitations: randomization was performed to determine the order of introduction of the intervention to clusters and not by individual participants. The ratings of the anxiety were recalled retrospectively and could have been affected by the arrival of the baby. Additionally, coping in labor was midwife assessed, which could have led to potential bias. The numbers within the subgroup analysis were small and this may perhaps have generated a spurious correlation. However, it is beneficial to note that subgroup analysis of treatment effects in subgroups of participants may provide useful information for the specific care of women and for future research.

## 5. Conclusions

Our results suggest that midwives providing an orientation on continuous labor support lowers women’s anxiety and enhances their ability to cope with pain during early active labor. Findings from this study may be of benefit in informing the development of protocols for the implementation of the presence of birth companions in similar low-resource settings. Future evaluation of the intervention is necessary to assess the effect of additional orientation during the antenatal period. An evaluation of the acceptability and perceptions of midwives regarding guiding or orienting birth companions is essential for implementation. Women’s experience of care during childbirth is key, and organized involvement of birth companions could improve women’s experience of care and consequently their health and future reproductive decisions.

## Figures and Tables

**Figure 1 ijerph-20-01549-f001:**
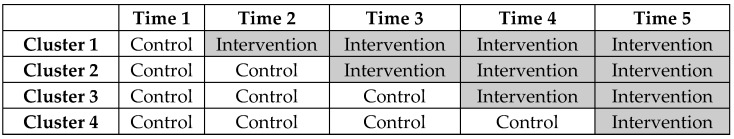
Trial design.

**Figure 2 ijerph-20-01549-f002:**
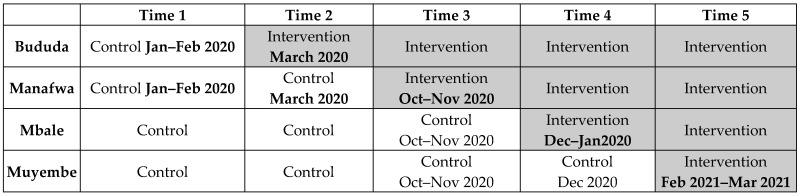
Trial duration and dates of study periods.

**Figure 3 ijerph-20-01549-f003:**
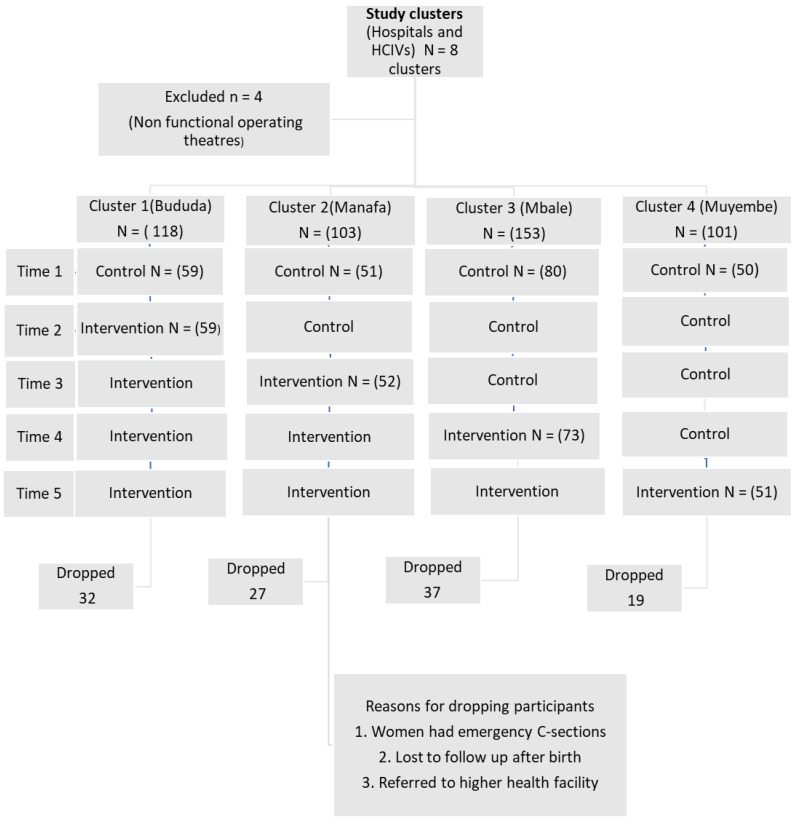
Trial flow chart.

**Figure 4 ijerph-20-01549-f004:**
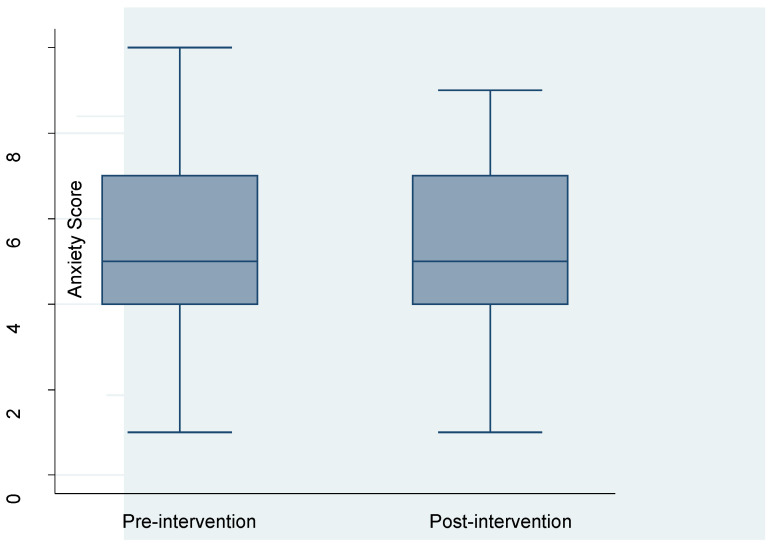
Anxiety score by study period.

**Table 1 ijerph-20-01549-t001:** Participant characteristics.

Characteristic	All Participants	Control	Intervention	*p*-Value
Age ^3^				
15–24	291(61.7%)	50.8%	49.2%	0.721
25–40	181(38.3%)	49.2%	50.8%	
Education				
Primary	252(53.1%)	52.4%	47.6%	0.544
Secondary	186(39.2)	49.5%	50.5%	
Tertiary	37(7.8%)	43.2%	56.8%	
Marital status ^10^				
Unmarried	99(21.3%)	29.3%	70.7%	0.000
Married	366(78.7%)	56.3%	43.7%	
Support person ^14^				
Spouse	155(33.6%)	60%	40%	0.003
Parent	185(40.1%)	46.5%	53.5%	
Sibling	103(22.3%)	36.9%	63.1%	
Friend	18(3.9%)	55.6%	44.4%	
Parity				
One	212(44.7%)	51.9%	48.1%	0.684
Two	109(23%)	45.9%	54.1%	
Three	83(17.5%)	50.6%	49.4%	
Four or more	70(14.8)	54.3%	35.7%	
Gestation weeks				
Mean (SD)	38.3	38.2 (1.0)	38.3 (1.0)	0.277
Cervical dilatation on admision				
4 cm	160(33.7%)	40%	60%	0.001
5 cm	113(23.8%)	50.4%	49.6%	
6–7 cm	201(42.4%)	59.2%	40.8%	
Birthweight Mean	3.2	3.2	3.2	0.819
Cluster Bududa	118	50%	50%	0.962
Manafwa	103	49.5%	50.5%	
Mbale	153	52.3%	47.7%	
Muyembe	101	49.5%	50.5%	

^3,10,14^ Number with missing data. Percentages compared by study arm.

**Table 2 ijerph-20-01549-t002:** Subgroup analysis; treatment effect on anxiety for specific baseline characteristics.

Characteristic	Control Mean (SD)	Intervention Mean (SD)	Diff.	*p*-Value
Facility Mbale Bududa Manafa Muyembe_Sironko	6.3(2.3) 5.9(1.3) 5.3(1.8) 5.0(1.7)	3.4(2.2) 6.4(1.3) 5.4(1.4) 5.4(1.3)	2.9 −0.5 −0.1 −0.4	0.704 >0.999 0.077 0.062
Age ^3^ 15–24 25–40	5.8(1.9) 5.6(1.9)	5.1(2.1) 5.2(1.9)	0.7 0.4	0.229 0.999
Education Primary Secondary Above secondary	5.9(1.9) 5.5(1.9) 5.9(2.2)	5.1(2.1) 5.2(2.0) 5.4(1.9)	0.8 0.3 0.5	0.565 0.625 0.532
Marital status Unmarried Married	6.1(1.7) 5.6(1.9)	5.9(1.8) 4.9(2.0)	0.2 0.7	0.766 0.488
Support person Spouse Parent Sibling Friend/relative	5.5(1.7) 5.8(1.9) 6.2(2.0) 6.5(1.8)	5.6(1.5) 5.1(2.1) 4.8(2.3) 5.5(1.8)	−0.1 0.7 1.4 1.0	0.331 0.345 0.361 >0.999
Parity One Two Three Four or more	6.1(2.0) 5.1(1.8) 5.0(1.5) 6.0(2.1)	5.3(2.2) 5.1(2.0) 4.8(2.0) 4.9(2.0)	0.8 0.0 0.2 1.1	0.329 0.451 0.070 0.787
Cervical dilatation on admission 4 cm 5 cm 6–7 cm	5.9(2.0) 6.2(2.0) 5.5(1.8)	5.3(2.1) 5.1(2.0) 5.1(2.1)	0.6 1.1 0.4	0.685 >0.999 0.126
Augmentation No Yes	5.6(1.9) 6.3(1.9)	5.1(2.0) 5.1(2.4)	0.5 1.2	0.470 0.217

^3^ Number with missing data.

**Table 3 ijerph-20-01549-t003:** Multivariable analysis of the effect of continuous labor support on maternal anxiety.

	Coef.	*p*-Value	95%CI
Control period Intervention period	−0.62	0.001 *	[−1.0–(−)0.2]
Age 25 and above	0.20	0.427	[−0.3–0.7]
Parity Two Three Four+	−0.41 −0.88 −0.32	0.103 0.003 0.354	[−0.9–0.1] [−1.5–(−)0.3] [−1.0–0.4]
Cervical dilatation on admission 5 cm 6 cm	−0.14 −0.88	0.191 0.003	[−0.5–0.5] [−0.7–0.2]
Support person Parent Sibling Friend/other relatives	−0.16 −0.34 0.50	0.479 0.191 0.275	[−0.6–0.3] [−0.8–0.2] [−0.4–1.5]

* Denotes statistical significance.

**Table 4 ijerph-20-01549-t004:** Proportions of women coping during different phases of labor.

	Control	Intervention	* *p*-Value
Coping at 4–7 cm Not coping Coping	41 (17.1) 199 (82.9)	24 (10.3) 210 (89.7)	0.031 *
Coping at 8–10 cm Not coping Coping Not applicable (Had cesarean section)	110 (45.8) 129 (53.8) 01 (0.4)	99 (42.3) 134 (57.3) 01 (0.4)	0.729
Coping 2nd stage Not coping Coping Not applicable (Had ceserean section)	95 (39.6) 125 (52.1) 20 (8.3)	83 (35.5%) 125 (53.4) 26 (11.1)	0.469

* Denotes statistical significance.

**Table 5 ijerph-20-01549-t005:** Showing treatment effect on coping for specific baseline characteristics.

Characteristic	Control-Total Number. (Proportion Coping)	Intervention-Total Number. (Proportion Coping)	Diff.	Prtest: *p*-Value
**Facility** Mbale Bududa Manafa Muyembe_Sironko	80(0.75) 59(0.79) 51(0.94) 50(0.88)	72(0.96) 59(0.78) 52(0.96) 51(0.88)	0.21 −0.1 0.02 0.0	<0.001 * 0.895 0.641 1.000
**Age ^3^** 15–24 25–40	148(0.81) 89(0.87)	143(0.89) 92(0.91)	0.8 0.4	0.057 0.389
**Education** Primary Secondary Above secondary	132(0.83) 92(0.84) 16(0.81)	119(0.91) 94(0.89) 21(0.86)	0.08 0.05 0.05	0.062 0.318 0.682
**Marital status** Unmarried Married	29(0.76) 206(0.86)	69(0.86) 160(0.91)	0.1 0.06	0.229 0.083
**Support person**				
Spouse	93(0.84)	62(0.90)	0.06	0.286
Parent	86(0.84)	98(0.89)	0.05	0.320
Sibling	38(0.84)	65(0.89)	0.05	0.464
Friend/relative	10(0.7)	8(0.88)	0.18	0.360
**Parity**				
One	110(0.79)	101(0.86)	0.1	0.049 *
Two	50(0.80)	59(0.92)	0.12	0.068
Three	42(0.95)	41(0.93)	0.02	0.701
Four or more	38(0.84)	32(0.97)	0.13	0.072
**Cervical dilatation on admission** 4 cm 5 cm 6–7 cm	64(0.81) 57(0.77) 119(0.87)	96(0.90) 56(0.86) 82(0.91)	0.10 0.22 0.380	0.109 0.218 0.379

* Denotes statistical significance; ^3^ Number with missing data.

**Table 6 ijerph-20-01549-t006:** Multivariable analysis of coping with labor.

	Unadjusted OR (95%CI)	*p*-Value	Adjusted OR (95%CI)	*p*-Value
Control period Intervention period	1.8 [1.1–3.1]	0.032 *	1.8 [1.0–3.2]	0.052
Age 15–24 25+	1.4 [0.8–2.5]	0.206	1.0 [0.5–2.3]	0.905
Parity Two Three Four+	1.3 [1.3–2.5] 3.3 [1.3–8.8] 1.9 [0.8–4.5]	0.387 0.160 0.138	1.3 [0.6–2.7] 3.3 [1.1–9.6] 2.3 [0.7–7.2]	0.460 0.033 0.166
Cervical dilatation on admission 5 cm 6 cm	0.7 [0.3–1.3] 1.2 [0.6–2.2]	0.228 0.612	0.6 [0.3–1.2] 1.1 [0.5–2.1]	0.150 0.861
Support person Parent Sibling Friend/other relatives	[0.6–2.0] 1.1 [0.5–2.3] 0.5 [0.2–1.8]	0.891 0.829 0.328	1.1 [0.6–2.3] 1.2 [0.5–2.6] 0.5 [0.1–1.7]	0.710 0.721 0.249

* Denotes statistical significance.

## Data Availability

Data are available on request from the corresponding author.

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
