# Peer review of "Effect of Midwife-Provided Orientation of Birth Companions on Maternal Anxiety and Coping during Labor: A Stepped Wedge Cluster Randomized Control Trial in Eastern Uganda"

_ijerph, 2023, doi:10.3390/ijerph20021549_

Round 1
Reviewer 1 Report
Thank you for the opportunity to review this paper.
The paper reports on an important trial and it is important that the results are published. The paper in its current form does not meet many of the standards required for the reporting of stepped wedge cluster randomised trials. Some of these will be easily corrected and are a matter of adding additional details.
The rationale for separately reporting the two outcomes of maternal anxiety and midwife reported coping are not provided. I remain unclear why these outcomes are not being reported in the main trial paper, and currently this is unjustified in the manuscript.
There are many issues with the current manuscript including:
1. The intervention is not correctly described. As continuous support was provided in both groups the intervention is ‘Midwife provided orientation’ and it needs to be described as such in the title and manuscript.
2. The intervention is poorly described. It states that one midwife provided all orientation, but it is unclear how this occurred in practice.
3. The baseline rate of spontaneous vaginal delivery that informed the sample size calculation needs to be included.
4. Although the units were randomised it appears that individual women were then recruited. This needs to be clarified and justified.
5. The proportion of eligible women recruited needs to be described.
6. The purpose of Table 1 is to be able to compare the characteristics of women in the control and intervention groups. Currently this is not possible as the characteristics between groups, rather than within groups, is presented.
7. The duration and dates of study periods needs to be described.
8. Maternal anxiety was captured relating to three periods, 4-7cm; 8-10cm and second stage, it is unclear which is being reported in Figure 3.
9. Justification for removing women who had a caesarean section needs to be provided. As the primary outcome of the trial was mode of birth, this cannot be justified.
1By convention, units would not normally be named.
1 A reduction in anxiety scores was seen in one unit with an increase in the other three units. This needs to be included in the discussion.
1 ‘Coping’ in labour was midwife, not women assessed. As the trial was unblinded this could lead to potential bias. This needs to be included in a section on weaknesses of the study.
This manuscript currently adds little to what would be expected to be the main trial paper. I would strongly suggest producing a single paper, of high quality, with all trial results rather than sub dividing results and performing additional secondary analysis.
Reviewer 2 Report
Dear author's
The subject is interesting .
Please explain the novelty of your study.
It is very important to explain the limitation of the study.
I suggest you to remove the sentence "This study aimed to assess the effect of continuous labor support from guided birth 338 companions on maternal anxiety and coping during labor" section conclusion.
If we talk about coping strategies I recommend you to read/cite the article "https://www.mdpi.com/2227-9032/9/4/466"In order to overcome a stressful situation, people use a variety of coping styles. Problem and emotional focused strategies are two main coping strategies described in various studies. Emotion-focused strategies have the purpose of regulating negative emotions and diminishing stress; examples of emotion-focused strategies include escaping and avoidance. In contrast, problem-focused strategies involve dealing directly with stressful situations, seeking support and information. Many studies focus on adaptive versus maladaptive coping strategies. Consequently, problem-focused coping is considered an adaptive coping style, whereas avoidant coping is deemed maladaptive—being associated with anxiety and, in some cases, with negative health behaviors that can also affect reproduction (smoking, drinking alcohol and taking drugs, poor sleep, weight issues).
Round 2
Reviewer 1 Report
Thank you to the authors for making the requested amendments to this important paper. I am very supportive of getting this paper published but it still needs a lot of work.
Apologies if I have noticed some more required corrections this time around. Although experienced in trials, I am not a statistician and given the methods and multiple subgroup analysis would suggest review by one. I have not seen a protocol and one is not lodged with the trial registry. As such it is unclear if the subgroup analysis now presented was pre-planned, or if what is presented are selective, post-hoc positive findings. If the analysis presented were not planned a priori this needs to be explained with the usual caveats.
Table 1 still does not present all the required data, and some are incorrect. The proportions for each subgroup within each characteristic should equate to 100%, currently many do not. For example, the totals provided for parity in the control group currently total 208%. I would suggest Table 1 is completely reworked. Characteristics that were different between the control and intervention groups needs to be acknowledged (not tested) and included in the multivariate analysis as confounders.
As above the proportions admitted at various cervical dilations is currently incorrect but once corrected the difference in cervical dilation on admission is potentially very important. Women in the control arm were admitted at a later stage in later in labour (4cm Control 26%, Intervention 40%; 6-7com Control 49.5%, Intervention 32.4%). This is understandable as it would be easier to recruit women to the intervention who present earlier in labour. The reason this difference is so important is that coping in ‘early active labour’ was recorded at 4-7cm. If this was recorded close to admission, women in the control group, admitted later in labour would have been assessed as ‘coping’ at a later point in their labour, and this may, at least partly account for the finding that women in the intervention groups coped better at 4-7cm. This at least needs to be acknowledged and cervical dilatation on admission included as a confounder (if it is not already).
The numbers of women ‘dropped’ needs to be provided for each reason. I would expect the women who had a caesarean section to have remained in the analysis until the point they needed to be excluded.
Given the potentially small numbers in the multiple sub-group analysis, n numbers need to be added and the potential for spurious results needs to be acknowledged.
Table 5 needs to include ‘coping at 4-7cm’ in the heading. It is also unclear why cervical dilation on admission is not included and described in Table 5.
There is an error in line 311 which currently reads:
Women in the intervention period had 80% higher odds 311 of coping (Unadj.1.8;95%CI 1.1-3.1) than those in the intervention period. Again if this refers only to coping at 4-7cm this needs to be made explicit.
Line 314:
‘This was the same even after adjusting for confounders’.
The confounders need to be listed.
In Table 6 and associated text it needs to be stated exactly what characteristics were adjusted for. It is unclear if this includes cervical dilation on admission. If it does not, it needs to be, given the later admission of women in the control group.
I am very supportive of getting this paper published but it must be scientifically sound.
Reviewer 2 Report
Dear author’s
I was read your revised paper.
We cant generalised your results because
the study is not randomised. This is a limitation of your study.
Otherwise the paper is relatively well written.
Please add a short conclusion with measures of improuving women health knowing the results of your study.
